# Tropical Monsoon Forest Thermodynamics Based on Remote Sensing Data

**DOI:** 10.3390/e22111226

**Published:** 2020-10-28

**Authors:** Robert Sandlersky

**Affiliations:** A.N. Severtsov Institute of Ecology and Evolution of the Russian Academy of Sciences, Russian-Vietnamese Tropical Research and Technology Centre, Leninsky Prospect 33, 119071 Moscow, Russia; srobert_landy@mail.ru

**Keywords:** exergy, entropy, information, non-equilibrium, self-organization, ecosystem, biological production, local climate, succession, order parameters

## Abstract

This paper addresses thermodynamic variables that characterize the energy balance and structure of the solar energy transformation by the ecosystems of deciduous tropical forests. By analyzing the seasonal dynamics of these variables, two main states of the thermodynamic system are determined: the end of the drought season and the end of the wet season. Two sub-systems of solar energy transformation are also defined: a balance system that is responsible for the moisture transportation between the ecosystem and atmosphere; and a structural bioproductional system responsible for biological productivity. Several types of thermodynamic systems are determined based on the ratio between the invariants of the variables. They match the main classes of the landscape cover. A seasonal change of thermodynamic variables for different types of thermodynamic systems is additionally studied. The study reveals that temperature above the forest ecosystems is about 4° lower than above the open areas during most of the year.

## 1. Introduction

Attempts to describe the work of living matter from the perspective of physics date back to the beginning of the 20th century. So, in 1925, Alfred Lotka, in *Elements of Physical Biology*, formulated an idea of living as a system different from nonliving by the properties of «internal environment» and an ability to sustain them with available energy [1]. Erwin Schrodinger, in 1944, [2] was among the first scholars to develop the idea that now became common: the living matter sustains its order (structure, organization) via the order extraction, or negative entropy, from the environment. An important aspect of “What is Life?” [2] ascertains the necessity of describing living matter or matter containing it from the point of thermostatic and thermodynamic models that should serve the comprehension of formation mechanics of chaos order, thus formulating the main goal of synergy. At the same time, Russian biologist Erwin Bauer [3] proved/argued that on the molecular level, life matter possess free energy to maintain a nonequilibrium state. In his concept of the biosphere, Vladimir Vernadsky [4] defined living matter as a transformer of solar energy and a source of free energy throughout the biosphere. During the second half of the 20th century, owing to the works of Brussels School [5,6,7,8], a concept of dissipative structures was formed, allowing to define living matter and biosphere in general as an open non-equilibrium system that supports its organization by solar energy influx and withdraws entropy excess into the environment.

In the second half of the 20th century, the thermodynamic approach began to be actively used in studies of various aspects of ecosystem functioning. Entropy naturally became the key parameter in the studies of living matter from the standpoint of thermodynamics [9,10,11], and then, with the appearance of concepts of energy quality in thermodynamics, exergy (free energy) follows [12]. Accordingly, the main subjects of such studies were the assessment of thermodynamic parameters for various objects—from individual organisms and even organs [13,14,15] to ecosystems and the biosphere as a whole [16,17,18]. Accordingly, the main problem of such studies was to test hypotheses about minimizing or maximizing the production of entropy in the course of both individual development and evolution/succession [19,20,21]. Based on the analysis of the literature, we distinguish three main levels of such studies in accordance with their objects:Biophysical (up to individual organisms). A comprehensive overview of thermodynamic applications in this area is given in Mustafa Ozilgen’s *Review on biothermodynamics applications* [22].Ecosystem (the level of communities and food chains). For a detailed overview, see *Thermodynamics in ecology* by Soren Nielsen with coauthors [23].Geophysical (land cover level). This direction, on the one hand, can be attributed to works related to the interaction of vegetation cover and climate [24,25,26], and on the other hand, research in which thermodynamic parameters are considered as integral indicators of the effectiveness of functioning [27,28,29], health [30,31], integrity/complexity/self-organization of ecosystems [32,33,34].

Naturally, the last two directions are characterized by a significant variety of thermodynamic parameters and methods of their measurement, determined by the specifics of each object. At the same time, according to Sven Jorgensen—one of the main ideologists of the thermodynamic approach in ecology [35], a complete assessment of the thermodynamic parameters of the ecosystem, and exergy in particular, is an impossible task so far. At the same time, the development of scientific tools gives researchers new instruments for measuring thermodynamic parameters. In the last decade, the Eddy covariance biophysical complexes of the Fluxnet network was a basis for thermodynamic assessment and modeling at the landscape cover level [29,36,37,38]. However, despite high measurement frequency, these complexes are representative only for local conditions (up to 0.5 × 0.5 km) and, at best, characterize a type of system in which they are installed. They can be supplemented by remote sensing data, which makes it possible to simultaneously assess the state of many ecosystems.

The introduction of multispectral remote sensing data drastically enlarged the possibilities for environmental scientists. However, the fundamental science was methodologically unprepared to use multispectral data as direct measurements of biophysical parameters of ecosystems. The functioning of the ecosystem (energy transformation) was assessed by using solely thermal bands [39,40,41,42,43], while data from the visible range, if applied, was used only to assess the condition of vegetation cover. One of the main problems that were restraining the development of “biophysical interpretation” of multispectral measurements was possibly a lack of theoretical and methodological basis that would have allowed surpassing traditional climate models, which ignored multispectral changes due to the lack of covered space. This basis was presented for the first time by two outstanding modelers Sven Jorgensen and Yuriy Svirezhev in their monograph *Towards a thermodynamic theory for ecological systems* [35].

Later, the scientific team under the leadership of Yuriy Puzachenko adapted this methodology for multispectral Landsat images and applied it to the study of landscapes of southern taiga in Central Forest Reserve [44,45,46], where a massive pool of field data on ecosystem properties and a functioning eddy covariance flux Tower, specializing in pulsating measurements of heat and carbon, already existed. The methodology was later adapted for Terra MODIS satellite imagery and used to analyze thermodynamic systems of both biosphere in general [47,48] and landscapes of the East European plain [49]. We see the following development in this area of study, on the one hand, in the more complex non-extensive Tsallis thermodynamics models (test calculations were completed for southern taiga landscapes of Central Forest National Reserve and the Russian Plain) [50]. On the other hand, we believe that a set of investigated biomes should be expanded. Thus, it is a particular scientific interest to us to compare thermodynamic systems of well-studied boreal and tropical forests. Ultimately, the combination of spatial multispectral data with high-frequency measurements by Eddy covariance should create a new basis for assessing the contribution of ecosystem structure to climate formation and contribute to a better understanding of vegetation modes of operation at different spatial scales. We believe that the implementation of this approach is possible based on a combination of the approaches of non-extensive non-equilibrium thermodynamics, information theory and synergetics. Regrettably, the information-thermodynamic analysis of ecosystems based on multispectral remote sensing information, the foundations of which were laid 20 years ago [51], has not yet received wide recognition. Thus, although in the monumental review of thermodynamics applications in the studies of ecosystems work *Thermodynamics in Ecology* [23], our esteemed colleague Soren Nielsen with coauthors reviewed more than 300 works in this area, they did not cite a single one where multispectral data were used as tools for direct calculation of thermodynamic parameters. Accordingly, in this work, we tried to expand the geography of the application of our information-thermodynamic analysis on the one hand. On the other hand, we tried to compare our earlier data for the forests of the boreal zone of the European Plain [46] with new data on the operation of tropical forest systems.

In the current study, we present our first results of the analysis of the functioning of thermodynamic system of the tropical deciduous forests in Southern Vietnam. To conduct a survey, we used the data collected by N.V. Sukachev Laboratory of biogeocoenology and Russian–Vietnamese Tropical Research and Technology Centre (Tropical Center) in Cát Tiên National Park. Since 2012, our laboratory has been measuring heat flux and greenhouse gases using an Eddy covariance method. Numerous zoological and botanical studies have been conducted in the Park since the late 1980s. In 2017, we began field research on the vegetation structure and components on a transect with a 20 m regular sampling step. Thus, by studying the thermodynamic system of Cat Tien using multispectral data, we can obtain a better understanding of the functioning of tropical ecosystems as regards perennial field observations.

## 2. Materials and Methods

Our research focused on the ecosystems of Cát Tiên National Park (11°26′ N, 107°24′ E), situated 150 km north-west of Ho Chi Minh (Figure 1a) and included the Dong Nai biosphere reserve. The study area lies in the tropical monsoon climate zone with a pronounced wet season from May to October and a drought period from November to April. The average annual precipitation is 2400–2500 mm/year. The average annual temperature is 26.4 °C, with December being the coldest month (23.9 °C) and April being the warmest month (29.1 °C). Monthly precipitation can reach 440 mm/month during the wet season and only 10–15 mm/month during the drought period. The relative humidity is approximately 70–75% during the drought season and 80–90% during the wet season [52]. The National Park is a relatively integrated massif of monsoon deciduous forest that occupies the southern spurs of the Central Plateau and The Dong Nai River valley (Figure 1b). Landscapes of the Park can be divided into three main types (generalized according to [53,54]): (1) flat plain with a basaltic base, partly covered by early Holocene tuffs under Lagerstroemia and Dipterocarpaceae vegetation and forests derivative from it on tropical reddish–brown tropical clayey soils with shallow profiles (the western part of the Park, 120–150 m above sea level); (2) undulating shale valley with Dipterocarpaceae vegetation and forests derivative from it on flat and protuberant surfaces and with bamboos on slopes with yellow ferralite or shallow tropical clayey soils (the eastern part of the Park, 300–350 m above sea level, the so-called Tiger Hills); (3) alluvial lake plains, occupied mostly by grasslands, wetlands and seasonally flooded low-trees forests on alluvial loamy soils (the north-western part of the Dong Nai River Valley and adjacent Lake Bau Sau basin). According to A.N. Kuznetsov and S.P. Kuznetsova [53], vegetation cover of Cat Tien was affected by defoliants during the Vietnam War (1961–1975). Since the end of war and until 1989, flat parts of the Park were subject to selection cutting. At the same time, some researchers [55,56] believe that the western flat part of the Park is occupied mostly by secondary forests on the former clearings. On the contrary to them, A.N. Kuznetsov and S.P. Kuznetsova argue that it was primarily young trees that were cut. Based on their other works on vegetation in Vietnam, the authors define Cat Tien forests as “primary valley seasonally flooded forests, effected by phytoxicants during the Vietnam War and selection cutting” [53] (p. 42). Literature data on the Orange Agent spraying [57,58] shows that almost the entire territory of Cat Tien was affected by it at least several times in 1967–1968. At the same time, the analysis of Landsat 5 and 7 satellite images with the 30 × 30 m spatial resolution for 1973–2016 [59,60] reveals that there were no significant disturbances or losses in biological productivity in the area (according to NDVI dynamics assessment) during the aforementioned period.

For the measurement system, we used the multispectral Landsat satellite imagery (Landsat 5 and 7) with the 30 × 30 m spatial resolution for the period between 2010 and 2013. We picked 17 images shot during the dry season from October to May with low cloudiness (parameters of the scenes are given in Table 1). The choice of the measuring system is defined by several reasons: an optimal set of spectral bands of the imaging system (visible, infrared, thermal), which quite fully reflect the work of the vegetation cover, a spatial resolution of 30 × 30 comparable to the size of an elementary community (crown of one large tree, for example), a large time series of data (1984–2013) and open access. Unfortunately, the measurement system of Landsat 8 differs from Landsat 5 and 7, as it has a different spectral bandwidth; thus, their joint analysis proves to be problematic for a biophysical approach. Therefore, we decided not to use this data together in this work.

Sensor calibration constants were used to recalculate brightness values of all channels, except the thermal band, into the radiation reflected by the active surface (a part of the surface that interacts with the atmosphere due to heat and moisture exchange) and are registered by sensors of the imagery system (W/m^2^). Brightness values of the thermal band were converted into the heat flux from the active surface and its temperature. The income of solar radiation in each spectral reflectance band was calculated using the solar constants for each band with respect to the solar altitude and the distance between the Earth and the Sun at the moment the image was taken. Accordingly, the energy absorbed by each band is considered as a difference between incoming and reflected solar radiation.

Thermodynamic properties for ecosystems are calculated according to the methodology of S.E. Jorgensen and Yu. M. Svirezhev [35] and corrected for the Landsat satellites [44,45,46]. In its most basic form, the energy balance of nonequilibrium system (B) includes exergy (EX), bound energy that cannot be converted into useful output (STW) and internal energy of the system (U):B = Ex + STW + U.(1)

As for the simultaneous assessment of thermodynamic variables during the single scene acquisition, it is more appropriate to use the increment of these values rather than absolute meanings. Exergy is the energy that can be converted into useful work. In the ecosystems, exergy is closely related to sustaining the water cycle. Bound energy is the energy that is dispersed into the environment together with heat flux and entropy. Internal energy increment means energy accumulation, probably in the form of organic matter increase. Exergy is evaluated as a function of non-equilibrium of incoming and reflected solar radiation specters (increment of Kullback information). The more the specters converge, the more equilibrium the ecosystem receptor reaches with the flux of incoming energy, hence the information increment is smaller. The increment of Kullback information for the Landsat satellite imagery (K, nit) is calculated as:K = ∑p*_i_^out^*lnp*_i_^out^*/p*_i_^in^*,(2)
where p*_i_^out^* = e*_i_^in^*/E*_in_* is the ratio of incoming energy (e*_i_^in^*) in the spectral band (*i*), and the total incoming energy (E*_in_*); p*_i_^out^* = e*_i_^out^*/E*_out_* is the ratio of reflected energy (e*_i_^out^*) in the spectral band (*i*) and total reflected energy (E*_out_*).

The exergy of solar radiation (Ex) is calculated as:Ex = E*_out_*(K + lnA) + B,(3)
where E_in_ is incoming solar radiation, W/m^2^; E*_out_* is reflected solar radiation, W/m^2^; B = E_in_ − E_out_ is absorbed energy; and E_out_/E_in_ is albedo.

To evaluate bound energy (energy dissipation with a heat flux and entropy), it is necessary to assess the entropy of reflected solar radiation. The larger is the entropy of the reflected solar radiation, the closer to equilibrium its flux is. The entropy (Sout, nit) is calculated as:S = −∑p*_i_^out^*lnp*_i_^out^*,(4)

Bound energy (STW, W/m^2^nit) is estimated as:STW = TW × S,(5)
where TW is the heat flux from the active surface, captured by a heat channel.

The increment of the internal energy of the system (DU) is a transition of absorbed solar energy into the internal energy of the system. It is estimated as a residue from the “balance” equation of absorbed energy (B):DU = B − Ex − STW,(6)

To assess the energy consumption for biological productivity, we used the ratio of standard channels: the difference between reflected energy in the red (RED) and short-range infrared ranges (NIR):VI = NIR − RED,(7)

Thus, in this study, we address a spatiotemporal variation of thermodynamic variables: the consumption of solar energy (W/m^2^), the information increment (nit), the entropy of reflected solar energy (nit), the heat flux from the active surface (W/m^2^), the exergy of solar radiation (W/m^2^), bound energy (W/m^2^nit) and the increment of internal energy (W/m^2^).

We also calculated the level of Foerster self-organization [61]:R = 1 − H/H_max_,(8)
where the system energy is H = −∑p*_i_*log2p*_i_*, H_max_ is the maximum entropy −log2p*_i_*k, where k is the number of discrete classes. Thus, the level of self-organization in the system increases when dR/dt > 0.

The spatiotemporal variation of thermodynamic variables is analyzed using the “ecosystem as a dynamic system” approach, where elementary elements are Landsat pixels. By studying the temporal dynamics of each element in the space of thermodynamic variables, we obtain their phasic trajectories that allow us to define the stable structures in a spatiotemporal aspect. To find and analyze these structures and the relations they cause, we used the synergy concept of order parameters (invariant state) and controlling parameters (external factors like climate and relief, or internal ones like self-development of the vegetation). In this analysis, the instruments of order parameters (invariants) were derived using the method of principal components. Invariants, defined by the method of principal components, are an orthogonal basis derived from the initial set of properties, in which the elements of a system (individual pixels) are located. According to the theory of dynamic systems, the combination of system elements with similar trajectories form attractors. Determined by the general mechanisms of regulation of the environment fluctuation, they can be associated with integrated spatial systems (plant communities at different levels). Groups of neighboring points with different trajectories form a spatiotemporal continuum that demonstrates the change of functioning in the invariant space. The ratio between these extreme forms determines the level of landscape organization. By classifying elements of the system (pixels) based on the combination of invariants, it is possible to determine groups of elements with similar trajectories (types of thermodynamic systems) and to assess the dynamics of the thermodynamic variables for these groups.

## 3. Results

Figure 2 shows the seasonal patterns of average values of thermodynamic variables for the territory depending on the incoming solar energy. During the seasons, the components of the absorbed energy balance (a) and the vegetation index (b) changed in a similar way. In general, structural and informational parameters illustrate the analogous dynamics: self-organization and information increment are increasing in May, and the entropy of the reflected solar radiation decrease. Regardless of the radiation income, the temperature of the active surface varies quite chaotically, possibly proving that local air masses influence it. The temperature may fall to 16–17 °C because of the haze that is not detected by sensors in the visible and near-infrared ranges. The average temperature does not rise above 24 °C even at the end of the drought season, when average monthly temperatures (according to the long-term measurements and Flux Tower results) reach 29 °C. This fact is probably determined by two circumstances. First, it is the relatively early time of image acquisition (around 10 a.m. local time). Second, unlike the Flux Tower, the measurements of the heat bands by Landsat satellites show the temperature field of the surface atmospheric layer that is significantly cooled by a powerful evaporation flux from the forest and not by the area above the tree crowns themselves. The latter process is described in the study [62], which compares the Landsat measurements and eddy covariance data in sphagnum–blueberry spruce forest.

In general, we can state that the studied thermodynamic system becomes particularly active with the income of solar energy and increasing precipitation. The input of the incoming solar energy (EIN) can be evaluated, basing on the regressive model for the energy expenditure on evapotranspiration (the exergy, EX):EX = 27.53 + 0.43EIN, (R^2^ = 0.77).(9)

The association between the vegetation index (VI) and the income of solar radiation:VI = −1.31 + 0.062, (R^2^ = 0.54).(10)

Structural and informational parameters and temperature do not depend on the income of solar energy. Our scientific experience on the analysis of the correlation between thermodynamic characteristics and the weather for boreal ecosystems during the vegetation period [46] shows that the variations of these parameters are related to the accumulated precipitation and temperatures. Unfortunately, the length of time series of Flux tower data (since 2012) does not yet allow us to assess their influence, but, as new satellite images and Flux data appear, the issue will eventually be solved. Another problem in the analysis of seasonal variations of thermodynamic variables in the monsoon climate is a lack of cloudless scenes during the wet period from June to August. Naturally, it complicates the assessment of the entire picture. In fact, we cannot evaluate one of the extreme conditions of the system: maximum moisture availability or the income of solar radiation.

To analyze spatiotemporal variation of thermodynamic variables for all 17 Landsat scenes, we unified each of them by using the principal component analysis. Table 2 shows the percentage of variation, which is characterized by the first two order parameters (principal component axes) for each thermodynamic variable, and the key scenes that define them. We also performed this analysis for all variables simultaneously using all scenes. It is clear from the table that, on average, spatiotemporal variation of each variable is described by two order parameters for 70–80%. For most variables, the first invariant determines their spatial variation at the end of the wet period inbetween September and February (March). These are the following variables: the absorbed solar radiation, the increment of the internal energy, the vegetation index, and information properties. The excluded variables are the evapotranspiration (exergy): its variation being mostly determined by the second invariant (6.69%) in these dates; and bound energy with temperature (heat flux): their first invariant determines their variation in the period between November and May. Accordingly, second invariants describe other dates at the end of the drought season. Thus, we can draw a conclusion that the thermodynamic system of the studied landscape can exist in two different states: the end of the wet season and the end of the drought season. Figure 3 shows the relation of invariants for the exergy and the vegetation index for two states of the system, depending on the invariant of the absorbed solar energy. Overall, we can state that this relation changes only in the local areas. We also suppose that the system state does not alter drastically during the period of the wet season that we cannot observe.

In order to complete the integrated analysis, we derive order parameters from all variables across all the scenes (9 variables for each of 17 scenes). The first two order parameters describe 45.66% and 19.65% of the total variation, respectively. The first invariant determine the variation of structural and informational characteristics and the vegetation index; the second one defines the components of energy balance. At this point, the analysis of factor loadings shows that none of the factors or their combinations describe the heat field (temperature) completely. The order parameters are shown in Figure 4. It illustrates that the most productive, self-organized, and non-equilibrium ecosystems occupy the upland slopes of the western exposition in the eastern part of the Reserve (Tiger Hills), and separate shale bars in the south-west on the opposite bank of the Dong Nai River. These ecosystems are bamboo formations with Dipterocarpaceae. The least productive, least equilibrium, and least self-organized ecosystems are those of grasslands around the Lake Bau Sau, agricultural lands and other non-forested anthropogenized lands outside of the Park on the left riverbank of the Dong Nai. Variables for the forests with predominant Lagerstroemia formations in the flat western part of the Park are close to the average values. The second invariant that shows the spatial variation of components of the absorbed solar energy (mainly exergy, the expenditure of energy on evaporation) is the largest for the water bodies (Lake Bau Sau, the Dong Nai River) and the erosion pattern of Tiger Hills. In addition, its values increase slightly in the seasonally flooded lowlands in the flat western part of the Park. Similar to the first-order parameter, the anthropogenized territories on the left Dong Nai riverbank have the minimum absorption and expenditure of energy on evapotranspiration.

Each pixel has a set of values of order parameters (invariants), therefore, it is possible to determine comparative types of the thermodynamic system in each pixel. Figure 5a illustrates an example of a dichotomous classification of the territory (the third level, i.e., eight classes) in the space of values of two main invariants that generalize variations of all variables. Average order parameter values for the derived types are shown in Figure 5b.

Based on the estimated relations of thermodynamic variables and our knowledge of the area’s landscape cover, we can define the following types of the thermodynamic system:(1)Agricultural lands (plantations of coffee, hevea, cashew, etc.) and settlements with a minimum level of self-organization, biological productivity, and energy expenditure on evapotranspiration;(2)Meadows, including pastures, and grasslands (Elephant grass *Pennisetum purpureum*) with a low level of self-organization, biological productivity, and energy expenditure on evapotranspiration;(3)Bamboo formations with a maximum level of biological productivity and self-organization, but with a low level of energy expenditure on evapotranspiration;(4)Young forests of elaborate composition with a medium level of biological productivity and self-organization, but with a high level of energy expenditure on evapotranspiration;(5)Forests with dominating Lagerstroemia formations with a medium level of biological productivity and self-organization, but with a high level of energy expenditure on evapotranspiration;(6)Forests with Dipterocarpaceae formations with a high level of self-organization, biological productivity, and energy expenditure on evapotranspiration;(7)Seasonally flooded areas occupied by shrubs and grasslands with a low level of biological productivity and self-organization, but with a high level of energy expenditure on evapotranspiration;(8)Water bodies (Lake Bau Sau, the Dong Nai River).

Overall, the defined thermodynamic systems correspond with the main formation types in the studied area. Moreover, their spatial configuration corresponds not only to our concept, but also to the results of other studies as well. For instance, in 1993, the landscape cover for Cat Tien National Park was classified with the use of the Landsat 5 satellite image from 1991 and verified later by geo-botanical field descriptions [55]. The map developed during that research generally corresponds to ours.

Figure 6 shows the thermodynamic system classes in the coordinates of invariants of main variables for two periods. As is evident from the figures, the types are more differentiated in terms of productivity and temperature at the end of the wet season, and in productivity and evapotranspiration at the end of the dry season. Understandably, the classes are more compact and discrete at the end of the dry season. In general, variables ratios for the classes are similar in both periods: the temperature rises and productivity decreases with an increase in openness (no forest); the evapotranspiration costs increase with an increase in the forest age (late succession stage), and productivity increases with a decrease in the forest age.

Figure 7 depicts the seasonal trajectories of thermodynamic variables for the defined types of thermodynamic systems. Since most of the balance components transform concurrently, in this case it is enough to study only the main ones: the energy expenditure on evaporation (a), temperature (b), and the vegetation index (c). Structural and informational characteristics also convert concurrently: the more the information increment is, the more significant the organization and the less the entropy of reflected solar radiation. Hence, we examine only the self-organization (d). The energy expenditure on evapotranspiration is the largest for the water bodies throughout the whole year, except on 11 May 2010 (131st day of the year). The following types are sorted by the exergy level in descending order: forests with Dipterocarpaceae formations, forests with Lagerstroemia formations, and young forests. Throughout most of the year, the evaporation costs in the seasonally flooded formations are similar to the young forests and bamboo formations; however, at the end of the wet season, when these territories are flooded, their values are close to the open areas. The energy expenditure on evaporation also increases in grasslands and agricultural lands at the end of the drought season. The annual change of spatial variation of the temperature field in open areas is quite insignificant; the highest temperatures are registered in the agricultural lands, followed by grasslands and seasonally flooded formations (shrubs and meadows). Forests, water bodies, and bamboo formations are even colder, with the latter being the coldest during certain periods. According to the vegetation index (c), the biological productivity is largest for the bamboo formations and young forests. Forests with Lagerstroemia and Dipterocarpaceae formations, grasslands, and agricultural lands rate next. There are almost no significant changes in productivity variations, except the productivity increase by the end of the year for grasslands and agricultural lands. Two groups stand out due to their values of self-organization (d): forests, including a bamboo formation with notably high results, and open areas with extremely low values. Overall, bamboo formations and young forests have maximum self-organization. However, forests with Lagerstroemia and Dipterocarpaceae formations have higher results during particular periods at the beginning of the wet season. Grasslands are the leaders in self-organization among the non-forested areas. At the end of the drought season, water bodies and agricultural lands have minimum self-organization. However, it increases at the end of the wet season for the agricultural lands and surpasses self-organization for the seasonally flooded formations.

## 4. Discussion

The energy expenditure on evapotranspiration generally increases together with the age of forest stand, while biological productivity does vice versa. The temperature variations are not entirely predictable, neither for the separate parts nor for the area in general. Thus, bamboo formations on the south-eastern slopes have a relatively low average temperature. Overall, the temperature increases with a decrease in tree cover. Figure 8 shows box-plots of the first temperature order parameter (the end of the dry season). Invariant’s values are converted to the values of 1 April 2007, which have the largest input (factor loading) according to the regression equation. According to the plot, the temperature fields of open areas (agricultural lands) are 4° warmer than those of the forests (25°–21°). We obtained the same patterns for the boreal landscapes of the Russian Plain [46]. This fact demonstrates a similar level of climate regulation in tropical deciduous and boreal mixed forests. 

There are two different states of the system in tropical forests and in boreal forests. However, in boreal forests, these states are determined by seasonal variation of the solar energy income and the ratio of the heat flux (temperature) from the active surface and the heat expenditure on evaporation (in winter, the expenditure and absorption of heat on evaporation increase when the heat flux is high; it is opposite in summer, when larger heat expenditure and absorption is accompanied by a lower heat flux). On the opposite the tropical forest does not demonstrate such a clear pattern: in general, the income of solar energy, its absorption or expenditure on evaporation does not strongly influence the temperature field. Contrary to boreal forests, the temperature field is mostly regulated by the local circulation and precipitations.

The analysis of the spatiotemporal variation of self-organization for the territory as a whole, based on the observation periods available to us, generally coincides with other results obtained from the Flux Tower data for tropical rain forests, for example, for the forests of the Central Amazon [63] and China [64]. These studies have shown that, like the general activity of forest vegetation [65], the self-organization increases during periods with the highest rainfall. Of course, the presence of the pronounced dry season for our territory should impose its own specifics. Thus, a study by Song with coauthors [66] shows a significant decrease in self-organization during the dry season. However, in the work of our colleagues, comparing the data of the Cat Tien Flux tower [67] with other measurement stations, it was shown that, according to the seasonal variation of the components of the energy balance, the studied area is closer to non-deciduous tropical forests. Apparently, this feature of the ecosystems of the study area is due to the unique soil and plant properties of the Cat Tien ecosystems. In the dry season, only individual trees of the first story and a part of the second story fall off (Lagerstroemia and Dipterocarpaceae families). The relatively flat relief of the part of the national park, where the Flux Tower is installed (Figure 1) and the peculiarities of tuff soils (a large number of pores), allow stretching the moisture reserves for almost the entire dry season.

If we consider the identified classes of the thermodynamic system as elements of a succession series, then the obtained relations fit well with the idea that as successional changes, free energy (exergy) increases, reaching a maximum in climax communities for tropical biomes [33,65]. We have shown the same result in a detailed analysis of the thermodynamic parameters of solar energy conversion for boreal ecosystems [46].

## 5. Conclusions

Our results correspond to the conception of the organization and functioning of the ecosystems of deciduous tropical forests. The defined types of thermodynamic systems are a strong basis for the future analysis of the differentiation of seasonal dynamics. These types can also be utilized when planning field research on the ecosystems’ characteristics that determine their differentiation. However, it is necessary to perform detailed studies on the relationship between the properties of forest ecosystems thermodynamic variables, the input of deciduous species in particular. We have already started to collect field data (LAI measurements, description of species, etc.) for this purpose. At the same time, the possible distortions due to the early time of actual scene acquisition must be considered when using the defined types of the thermodynamic systems. For instance, areas with maximum energy absorption and expenditure on evapotranspiration occupy the most shaded slopes of the north-western exposition. Thus, it is crucial to assess the input of relief into the thermodynamic properties for further research. This can be achieved by standard statistical methods of analyzing the digital elevation model and its derivative morphometric characteristics. 

The approach demonstrated in this work showed a substantial similarity of the results relative to the results obtained for other regions according to the data of another measurement system—eddy covariance. We observe a decrease in the activity of the tropical forest system during the wet season (a decrease in the production of entropy, self-organization and exergy) and a similar decrease in its succession range from anthropogenized communities to young regenerating forests to indigenous dipterocarp forests. The analysis of the data for the available measurement periods (without the wet season) showed the presence of two-phase states of the thermodynamic system of the monsoon deciduous tropical forest—the end of the wet season and the end of the dry season. As for boreal forests, significant independence of the subsystems of evaporation and biological production is shown. Unfortunately, so far, the length of the data series on our Flux Towers located in the boreal ecosystem (European part of Russia) and in the considered ecosystem of Cat Tien does not allow us to extract a dependence with Landsat estimates sufficient for statistical analysis. However, in the future, as data is accumulated, we will be able to move in this direction.

## Figures and Tables

**Figure 1 entropy-22-01226-f001:**
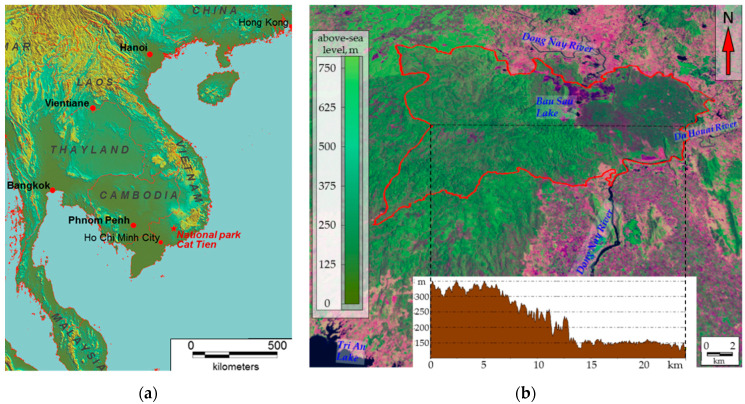
Study area: (**a**) Geographical position of Cat Tien National Park. (**b**) Cat Tien National Park, Landsat 8 image 04.04.2020 (false color image, spatial resolution 30 × 30 m) on digital elevation model SRTM.

**Figure 2 entropy-22-01226-f002:**
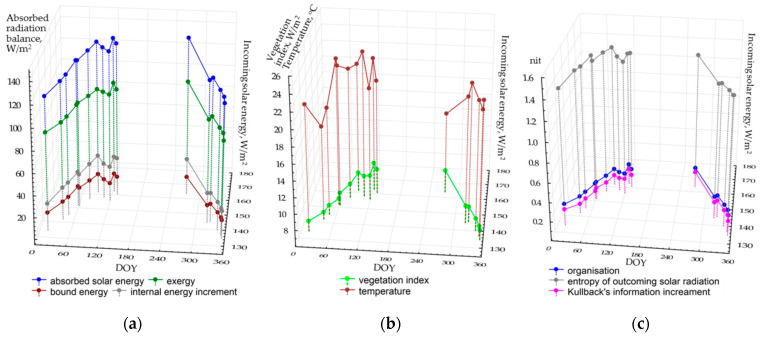
Seasonal variation of mean thermodynamic variables depending on incoming solar radiation: (**a**) balance components; (**b**) temperature and productivity; (**c**) structural and informational characteristics.

**Figure 3 entropy-22-01226-f003:**
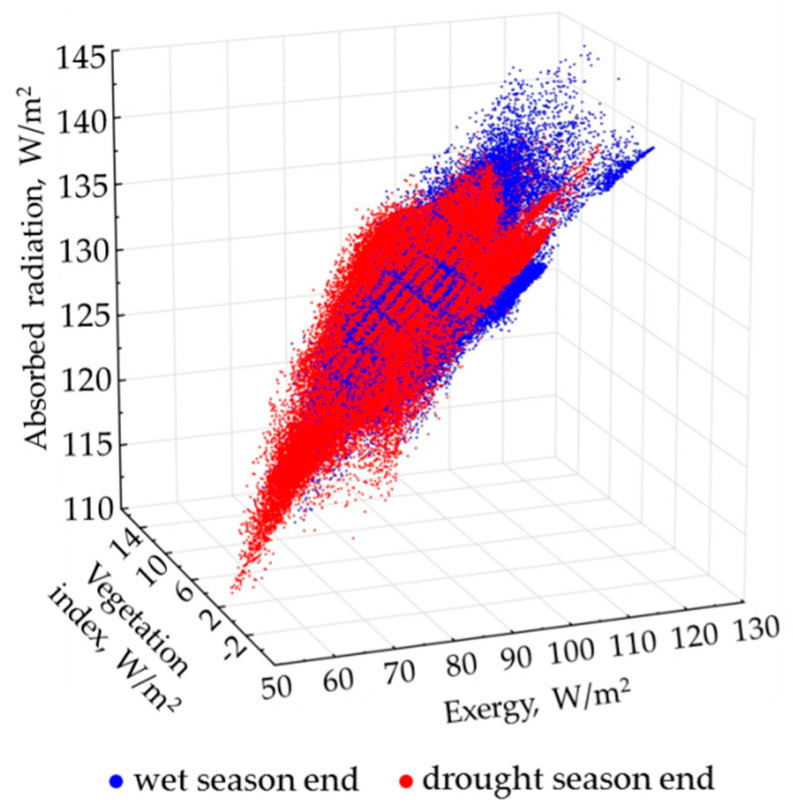
The ratio between order parameters (invariants) of absorbed solar energy, exergy, and vegetation index for the end of the dry and the end of the wet season.

**Figure 4 entropy-22-01226-f004:**
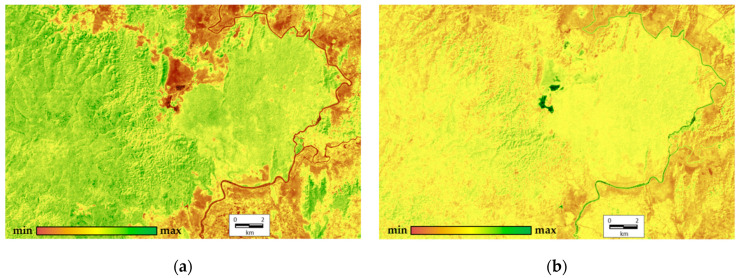
Order parameters (invariants) of thermodynamic system: (**a**) first order parameter (45.66%), structural and informational characteristics and vegetation index; (**b**) second order parameter (19.65%), absorption energy and exergy.

**Figure 5 entropy-22-01226-f005:**
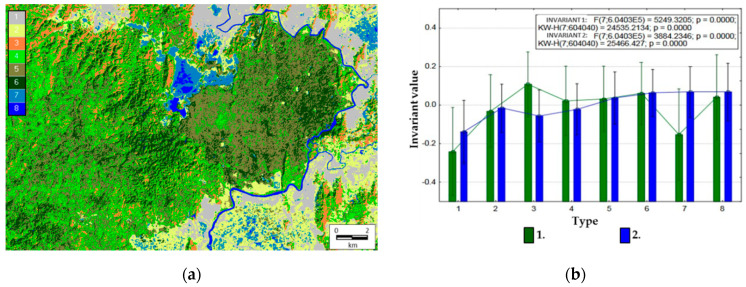
Types of thermodynamic systems: (**a**) dichotomic classification: 1, agricultural lands and settlements; 2, meadows and grasslands; 3, bamboos; 4, young forests; 5, Lagestroemia forests; 6, Dipterocarp forests; 7, seasonally flooded grasslands; 8, bodies of water. (**b**) Means for invariant values by types: 1, first invariant (structural information characteristics and vegetation index); 2, second invariant (absorbed energy and exergy).

**Figure 6 entropy-22-01226-f006:**
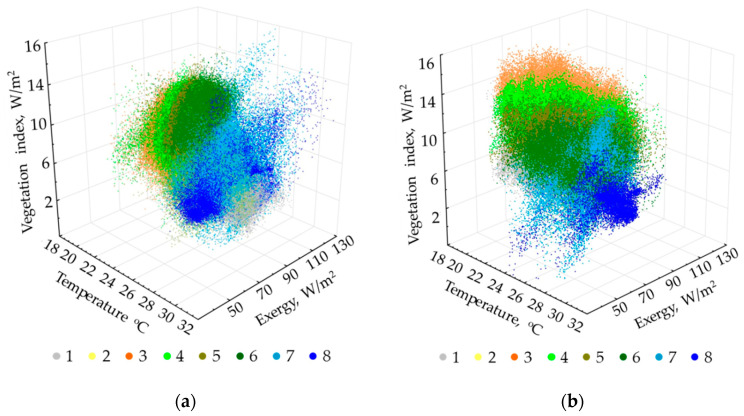
Arrangement of thermodynamic system types in space of order parameters of the main thermodynamic variables: (**a**) End of wet season; (**b**) end of drought season. Type numbers detailed in text and Figure 5 description.

**Figure 7 entropy-22-01226-f007:**
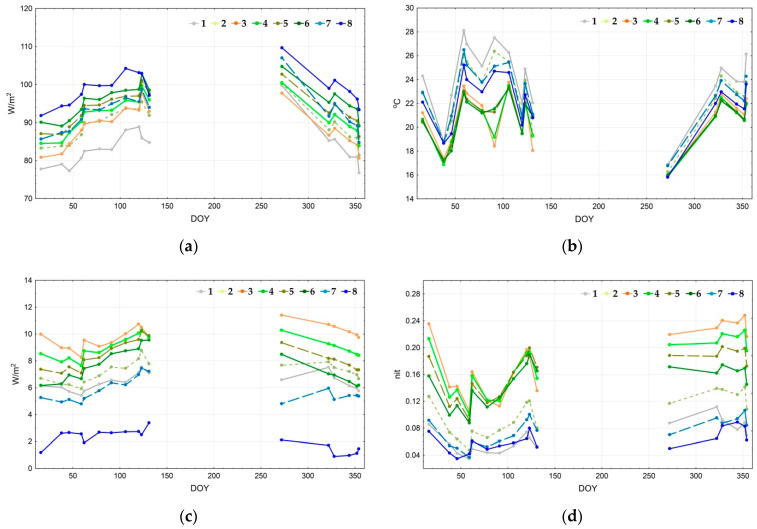
Seasonal dynamics of main thermodynamic variables for thermodynamic system types: (**a**) exergy (evapotranspiration); (**b**) temperature; (**c**) vegetation index; (**d**) self-organization.

**Figure 8 entropy-22-01226-f008:**
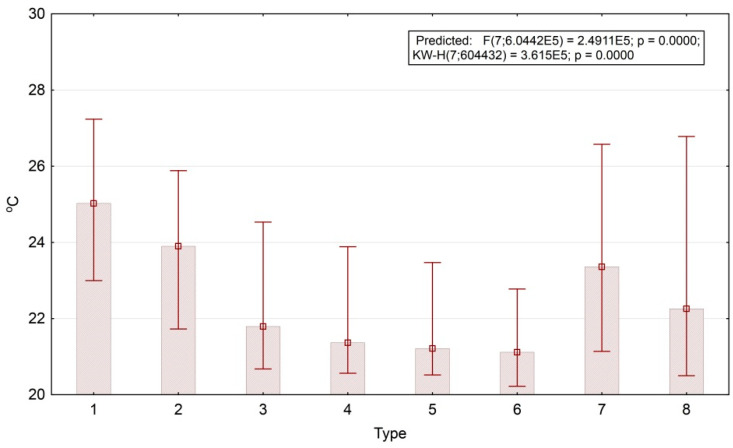
Temperature invariant for the dry season and mean values for thermodynamic system types. Type numbers detailed in text and Figure 5 description.

**Table 1 entropy-22-01226-t001:** Used Landsat scene parameters.

Sensor	Day of Year (DOY)	Date			Local Time	Sun Elevation ^o^	Incoming Energy, W/m^2^
		Day	Month	Year			
Landsat 5 TM	16	16	January	2009	9:52	42.22	134.24
	38	7	February	2011	9:57	48.92	141.84
	46	15		2008	9:58	50.6	145.01
	59	28		2007	10:02	54.41	151.69
	62	2	March	2008	9:57	54.22	151.15
	78	18		2008	9:57	57.93	156.55
	91	1	April	2007	10:02	61.7	161.41
	106	16		2001	9:47	60.34	157.91
	120	29		2000	9:43	59.87	156.05
	123	3	May	2007	10:02	64.51	162.61
	131	11		2010	9:58	63.51	160.6
Landsat 7 ETM+	272	28	September	2000	9:58	61.35	163.34
Landsat 5 TM	322	18	November	2004	9:52	50.37	144.26
	328	24		2006	10:02	50.57	145.01
	344	10	December	2006	10:02	47.83	139.86
	352	18		2009	9:58	46.36	136.72
	354	20		2004	9:53	45.42	134.66

**Table 2 entropy-22-01226-t002:** Order parameters (invariants) for thermodynamic variables obtained by PCA: the percentage of variation described by the invariant, and survey terms by factor loads that determine them.

Variables	Order Parameter 1		Order Parameter 2	
	%	Terms	%	Terms
Absorbed radiation	60.47	September–February	8.24	February–May
Exergy	65.73	February–May	6.69	September–February
Bound energy	78.39	November–May	4.03	May–November
Internal energy increment	58.42	September–February	7.35	February–May
Temperature (heat flux)	61.34	November–May	10.06	May–November
Vegetation index	67.58	September–February	9.43	March–May
Kullback information increment	72.68	September–February	9.43	February–May
Entropy of solar outcoming radiation	76.59	September–February	5.29	February–May
Self-organization	68.33	September–March	9.84	March–May
All variables	45.66		19.65	

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
