# Peer review of "Tropical Monsoon Forest Thermodynamics Based on Remote Sensing Data"

_entropy, 2020, doi:10.3390/e22111226_

Round 1
Reviewer 1 Report
The paper ideals with analysing vegetation and vegetation attributes by analysing satellite data. The paper is written in a good level of English, however it needs to be revised, to improve some sentences and correct spelling mistakes.
The paper has a long introduction including works from the first part of the 20th century, which are not giving relevant addition to the results of the paper. Also the literature review should be improved by addign more international research.
In Table 1 it is not clear which parameters can be understood as "science parameters". Please elaborate. Also in Table 1 all data points are from the same time appriximately (10 am). It would be necessary to justify the time selection and also to add why is it enough to only use these data?
The results presented in Figure 2 are hard to understand, these charts should be updated and better demonstrated (maybe from a different angle so the results are better shown).
On Figure 3 it would be good to add units to the axes as well. Also it is hard to see the results from this angle, e.g. it is not clear what is the relation of the two datasets where the absorbed energy is high.
On Figure 6 it is also hard to see how the different land covers are behaving. Aslo it would be good to describe what does the temperature parameter stand for on this Figure. Is it temperature difference?
On Figure 7 it would be better to break the connection over between May and September, since it is misleading. The same can be told for Figure 3.
On Figure 8 the line connecting the different types should be deleted, it has no purpose and is misleading.
Author Response
We deeply appreciate the kind review and appreciation of our study!
- Unfortunately, we are not native speakers, and the specificity of the text requires a highly qualified translation, which we did not dare to entrust to the translator. We have made some editing efforts.
- We've revised the introduction to match your recommendations. The literature review has been expanded and the historical part has been shortened.
- There is a typo in Table 1; instead of scientific parameters, read Landsat scene (survey) parameters. About the shooting time – the Landsat satellite is passing and, accordingly, shooting at specified location at a time which is defined by orbital parameters. We do not consider this time optimal, since according to the data of our colleagues from Flux-Tower, the maximum activity of the vegetation by evaporation (for example) is observed in 11-12 a.m. But we have no choice, we cannot influence the shooting time. Unfortunately, due specifics of analysis, we can only use a satellite system with one set of bands. Our calculations for other imaging systems naturally give different values ​​due to spectral bands widths. Even the Landsat 8 system gives different ratios than relatively close Landsat 5 (1984 - 2013) and 7 (1999 - 2003) and they cannot be used in the same time series. On the other hand, a set of spectral bands, a uniform 30x30m resolution comparable to the size of an elementary community (the crown of one large tree, for example), and a long history of surveying and free access make Landsat our only choice.
- We tried to remake the drawings in accordance with your comments: we changed the viewing angles and removed unnecessary lines. The figures, about which you had questions regarding the values of the axes, show the values of the order parameters (invariants) obtained by the principal component method. They are dimensionless. We reflected this in figures captions. All obtained parameters are positively related to variables, respectively, the greater the value of the order parameter (invariant), the greater the value of the particular variable.
Reviewer 2 Report
This paper addresses thermodynamic variables that characterize the energy balance and structure of the solar energy transformation by ecosystems in the deciduous tropical forests. In general, this paper is relatively well written. The topic is interesting and practical. However, attentions should be paid to several comments as follows:
- The research gaps between previous studies and your study should be offered clearly in the abstract and in the introduction.
- You should directly introduce your innovation in the introduction. It is very important for the authors to understand and support your study.
- The Literature reviews are poorly written, it is not a good idea to describe a document in too many sentences, why not classify these documents?
- A few more literature is needed in this paper. Only 27 literature is not enough.
- The figures should be more clearly. For example, the Figures 1, 2, 3 and 5 are blurred.
- What are your main conclusion? Please put them in the last part clearly.
Author Response
We deeply appreciate the kind review and understanding of our study!
We develop our approach alone and therefore the research as a whole is innovative. Nobody uses multispectral imagery to directly calculate the thermodynamic parameters of ecosystems / landscape cover. We rewrote the introduction to match your guidelines. We also added a literature review in the introduction and expanded the use of sources in the discussion of the results to show that, despite the novelty of the method, our results are quite consistent with others obtained by the more traditional ones (Eddy covariance). Ultimately, combining spatial (but isolated in time) assessments, with high frequency measurements on the Flux towers it will be possible to provide a more complete picture of system performance at any time and place. So far we are only on the way to this.
We also improved the quality of the figures.
Reviewer 3 Report
This article is about a very interesting topic. Broadly, I have no objections to the content of the article. However, I think the authors should check grammar and style. Please see the specific comments (and the text marked) on the word document. They should also check the bibliography. I recommend a proofreading service. Therefore, the article may be publishable. But first, the authors should make corrections.

Author Response
We deeply appreciate the kind review and understanding of our study! Also many thanks for the corrections and comments in the text of the article itself. Unfortunately, we are not native speakers, and the specificity of the text requires a highly qualified translation, which we did not dare to entrust to the translator. We have made some editing efforts.
Also, at the request of other editors, we have significantly changed the introduction, discussion and conclusion by adding a lot of references.
Round 2
Reviewer 1 Report
The English language has been slightly improved by the authors, which makes the level of English acceptable, however it could be further improved.
Figure 2 has been improved and now can be better understood.
Figure 3 has been enlarged, however the units for the "z" axis is still missing.
Figure 6 has been changed, however the requested additional information has not been added (e.g. the temperature and exergy base values)
Figure 7 and 8 has been improved and now acceptable.
Author Response
Thanks for the review.
We have changed the marked pictures.
Unfortunately, in the two days that the editors gave us this time, we have symbolically improved the language.

Reviewer 2 Report
Can be published
Author Response
Thanks for the review!
